# A Promising Biomolecule Able to Degrade Neutrophil Extracellular Traps: *Cdc*PDE, a Rattlesnake Phosphodiesterase

**DOI:** 10.3390/toxins15010044

**Published:** 2023-01-05

**Authors:** Isadora Oliveira, Victor Costa, Flávio Veras, Isabela Ferreira, Fernando Cunha, Thiago Cunha, Wuelton Monteiro, Eliane Arantes, Manuela Pucca

**Affiliations:** 1Department of BioMolecular Sciences, School of Pharmaceutical Sciences of Ribeirão Preto, University of São Paulo, Ribeirão Preto 14040-903, SP, Brazil; 2Department of Pharmacology, Ribeirão Preto Medical School, University of São Paulo, Ribeirão Preto 14049-900, SP, Brazil; 3School of Health Sciences, Amazonas State University, Manaus 69850-000, AM, Brazil; 4Department of Teaching and Research, Dr. Heitor Vieira Dourado Tropical Medicine Foundation, Manaus 69850-000, AM, Brazil; 5Medical School, Federal University of Roraima, Boa Vista 69310-000, RR, Brazil; 6Health Sciences Postgraduate Program, Federal University of Roraima, Boa Vista 69310-000, RR, Brazil

**Keywords:** phosphodiesterase, snake venom, *Cdc*PDE, svPDE

## Abstract

Neutrophil extracellular traps (NETs) are an important mechanism for defense against pathogens. Their overproduction can be harmful since excessive NET formation promotes inflammation and tissue damage in several diseases. Nucleases are capable to degrade NET on basis of their DNA hydrolysis activity, including the *Cdc*PDE, a nuclease isolated from *Crotalus durissus collilineatus* snake venom. Here, we report a new finding about *Cdc*PDE activity, demonstrating its efficiency in degrading cell-free DNA from NETs, being a potential candidate to assist in therapies targeting inflammatory diseases.

## 1. Introduction

Among the mechanisms involved in the immune system defense, neutrophil-derived extracellular traps (NETs) can be highlighted [1,2,3,4]. Indeed, NETs play an important role in controlling infections promoted by viruses, fungi, and bacteria [5]. NETs are composed of decondensed chromatin, and cytotoxic proteins, such as histones, neutrophil elastase, and myeloperoxidase [6,7]. Although they represent an important defense mechanism, NETs are highly encountered in the blood of patients during several inflammatory diseases, such as rheumatoid arthritis, psoriasis, systemic lupus erythematosus (SLE), COVID-19, and sepsis [8,9,10,11,12,13].

Venomous animal-derived toxins are known to display fabulous pharmacological properties, representing interesting lead compounds for the development of novel medicines. Indeed, some venom-derived drugs are already in the market, and several are under clinical trials [14]. The global pharmaceutical industry has benefited greatly from biodiversity-rich countries stimulating the bioprospection of novel biomolecules derived from venoms for novel drug design [15].

DNAses are potent agents capable of degrading NETs, being able to digest the DNA present in the mesh [16]. *Cdc*PDE, a phosphodiesterase, was recently (2021) isolated from the *Crotalus durissus collilineatus* rattlesnake venom, and was completely characterized. Moreover, functional previous studies demonstrated that the *Cdc*PDE was able to inhibit ADP-induced platelet aggregation and to cause a cytotoxic effect to human keratinocytes [17]. Knowing that *Cdc*PDE is classified as a nuclease, able to degrade DNA (i.e., DNAse [18,19], this study aimed to verify if the rattlesnake-derived molecule could degrade cell-free DNA from NETs (Figure 1).

## 2. Results and Discussion

The isolation and characterization of *Cdc*PDE was performed in our previous study, showing its low recovery (0.71%) [17]. *Cdc*PDE at 20 µg/mL (0.2 mM) exhibited significantly lower levels of cell-free-DNA in comparison with undigested groups (Figure 2), indicating its ability of cell-free DNA degradation. Although the highest tested concentration (20 μg/mL) inhibited ~38%, the amount of the used *Cdc*PDE was 25-times lower than the DNAse control (500 μg/mL). Unfortunately, we could not test *Cdc*PDE using higher concentration due to its very low recovery, making a dose-response curse unfeasible. Indeed, to enable to carry out a dose-response curve, assays with inhibitors, and in vivo tests, heterologous expression will be necessary.

Others substrates were tested in PDEs studies to characterize their enzymatic activity, such as *bis* (*p*-nitrophenyl) phosphate (0.4 mM [20], 1 mM [17] and 5 mM [21]), adenosine triphosphate (ATP, 0.05 mM), nicotinamide adenine dinucleotide (NAD, 0.05 mM), adenosine diphosphate (ADP, 0.05 mM), nicotinamide guanine dinucleotide (NGD, 0.05 mM), and adenosine monophosphate (AMP, 0.05 mM) [22], among others [18,19]. Although our study did not aim to characterize the enzymatic kinetics of *Cdc*PDE, using λDNA as substrate, we observed that the amount of substrate used in the cited studies is much higher than in our study (6 × 10^−13^ mM), and even though the hydrolysis was observed. Therefore, our study indicates that *Cdc*PDE may present a high specificity/affinity for the used substrate.

Only a few studies report that snake venoms can present DNAse activity [23,24]. Sittenfeld and colleagues (1991) tested DNAse activity of *Bothrops asper*, *B. godmani*, *B. schlegelii*, *B. lateralis*, *B. nasutus*, *C. durissus,* and *Lachesis muta* venoms, through radial diffusion in gel, and observed that all of them demonstrated DNAse activity [23]. Sales and Santoro (2008) have tested DNAse activity in 28 Brazilian venoms belonging to *Bothrops*, *Crotalus*, *Lachesis*, and *Micrurus* genera, observing that *B. brazili* presented the highest DNAse activity [24]. In contrast, here we pioneer demonstrated the DNAse activity of an isolated phosphodiesterase from the *C. d. collilineatus* venom. Moreover, to the best of our knowledge, this is the first study to test a snake venom-derived PDE using cell-free DNA from NETs.

Knowing that immune mechanisms participate on pathogenesis of several inflammatory diseases [25], the impairment of NETs degradation may promote endothelial damage, organ dysfunction, inflammation, and autoimmunity [26,27]. Supporting that, few studies showed that exogenous DNAse can improve the outcome of some diseases, such as sepsis and COVID-19 [28,29]. Therefore, attenuation of neutrophil-induced effects (i.e., NETs) may be a potent target for controlling diseases characterized by an influx of granulocytes and their activation, as was here demonstrated by *Cdc*PDE biomolecule.

## 3. Conclusions

Our communication supports the use of NETs’ inhibitors by degrading DNA-free from NETs, as a strategy to ameliorate multi-organ damage during the clinical course of NETs-associated inflammatory diseases. Although further studies are needed, our study pioneering shows that a snake-venom derived PDE can degrade cell-free DNA, which may contribute to reduce the pathogenicity of inflammatory diseases.

## 4. Methods

*Cdc*PDE was isolated from *Crotalus durissus collilineatus* in our previous study [17].

The DNAse activity was measured following the protocol of Colón and colleagues (2019) [29] with some modifications. Briefly, we diluted λDNA from the Quant-iT™ PicoGreen™ dsDNA Assay Kit (Cat. P11496, Lot. 2313058, ThermoFisher Scientific, Waltham, MA, USA) to a concentration of 20 µg/mL (1 µg = 0.03 pmol), the diluent was the Roswell Park Memorial Institute Medium (Ref. 15-040-CV, Lot. 17921005, Corning Inc., Corning, NY, USA). The λDNA solution was placed on a 96-well black plate with a clear bottom (Ref. 3603, Corning Inc., Corning, NY, USA), and after this, samples were treated with *Cdc*PDE (10 or 20 μg/mL), Pulmozyme™ (500 μg/mL, Roche, Basel, Switzerland) that was the positive control of DNAse activity, and PBS (Cat. 14190144, Corning Inc., Corning, NY, USA) that was the negative control of DNAse activity. On the same plate, we placed the samples, Pulmozyme, or PBS with RPMI medium without the λDNA solution, this well was used to correct the amount of DNA contained in the samples. We incubated the plate for 2 h at 37 °C and immediately after, we used the SYTOX Green™ (Cat. S7020, ThermoFisher Scientific, Waltham, MA, USA) to stain the remaining DNA in the wells. After incubation for 5 min protected from the light, we used the Flexstation 3 (Molecular Devices, San Jose, CA, USA) to read the fluorescence at 488 nm and excitation at 525 nm. Our results are shown as the percentage of fluorescence at each well. The average of the wells containing only medium and the λDNA solution was considered 100% fluorescence.

## Figures and Tables

**Figure 1 toxins-15-00044-f001:**
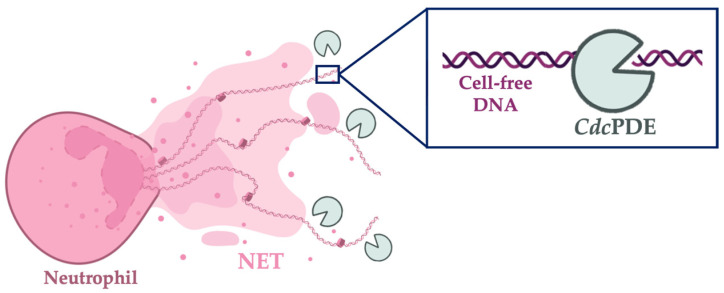
The mechanism of *Cdc*PDE-induced NETs’ degradation. A representative figure is used to illustrate the mechanism of NET degradation by a phosphodiesterase enzyme. The right panel shows the *Cdc*PDE degrading the presented DNA-free in the neutrophils’ mesh. The figure was created with BioRender.com.

**Figure 2 toxins-15-00044-f002:**
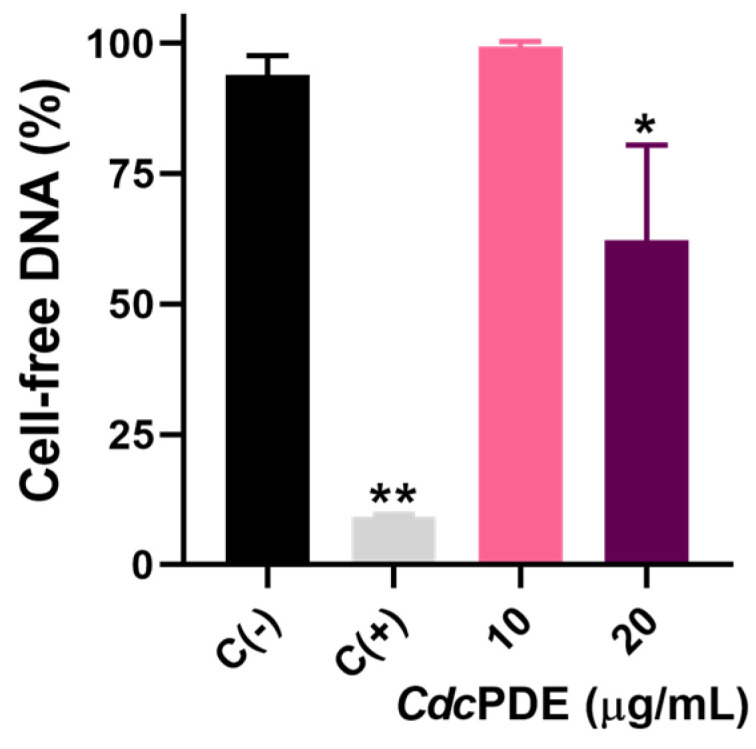
Cell-free DNA degradation assay. Isolated DNA were treated with PBS, DNase1 (500 μg/mL), or *Cdc*PDE (10 or 20 μg/mL) for 30 min at 37 °C. Wells were stained with 0.2 μM SYTOX Green for 10 min. Fluorescence emission intensities were determined using 488-nm excitation and 525-nm emission. The results are presented as relative fluorescence units (RFU) and bars express percentages relative to the PBS group. C (−): wells treated with PBS. C (+): wells treated with DNAse (500 μg/mL). * *p* < 0.01 and ** *p* < 0.0001 when compared with C (−). Data (n = 3) are presented as mean ± SD, which were analyzed by one-way ANOVA and Tukey’s post-hoc test using GraphPad Prism 9 software.

## Data Availability

Not applicable.

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
