# Peer review of "A Promising Biomolecule Able to Degrade Neutrophil Extracellular Traps: CdcPDE, a Rattlesnake Phosphodiesterase"

_toxins, 2023, doi:10.3390/toxins15010044_

Round 1

Reviewer 1 Report

This 'communication' manuscript reports a single finding: CdcPDE is able to degrade neutrophil extracellular traps. The finding is new and interesting and warrants publication.
However, some issus should be adressed before:
1) The authors might think about adding a sentence to the introduction to provide some guidance with enzyme nomenclature. Reading the short manuscript with different terms (phosphodiesteres, nuclease, DNAse) might be confusing.
2) The key experiment is visualized in figure 2. Do the error bars represent SD or SEM?
3) The most critical point is the usage of different concentrations (500 µg/ml vs. 20 µg/ml). The authors clarify why it was not possible to use a higher conentration of CdcPDE, but what about using a lower (and comparable concentration) for the control? This should be an easy task and would have the advantage to directly compare enzymatic activity.

Author Response

RESPONSE TO REVIEWERS

Title: A promising biomolecule able to degrade neutrophil extracellular traps: CdcPDE, a rattlesnake phosphodiesterase

Thank you very much for your considerable effort in reviewing our manuscript. It is also appreciated that you considered our work of interest for your journal and its readers to allow the submission of a revised version. It stimulated us to amend the text to meet your constructive comments. In what follows, you will find a point-by-point list of how we dealt with reviewer comments in blue, and necessary changes are highlighted in the manuscript in red. We hope that this version is now acceptable for publication in Toxins.

Comments of the Reviewers:

Reviewer 1:

This 'communication' manuscript reports a single finding: CdcPDE is able to degrade neutrophil extracellular traps. The finding is new and interesting and warrants publication.

However, some issus should be adressed before:

1) The authors might think about adding a sentence to the introduction to provide some guidance with enzyme nomenclature. Reading the short manuscript with different terms (phosphodiesteres, nuclease, DNAse) might be confusing.

Response: We added information in the manuscript as suggested. See lines 48-50.

2) The key experiment is visualized in figure 2. Do the error bars represent SD or SEM?

Response: The bars represent SD. The information was detail in the new version of the manuscript.

3) The most critical point is the usage of different concentrations (500 µg/ml vs. 20 µg/ml). The authors clarify why it was not possible to use a higher concentration of CdcPDE, but what about using a lower (and comparable concentration) for the control? This should be an easy task and would have the advantage to directly compare enzymatic activity. 

Response: We used a lower concentration of CdcPDE (10 µg/ml), and unfortunately did not see any significant result (see figure 2). Regarding the concentration of the control (+), it is standardized in the lab for the NET assay, a concentration able to degrade more than 90% of NETs.

Reviewer 2 Report

The manuscript is important and describes the action of a recently isolated nuclease. The authors showed that this enzyme cleaves DNA from cell-free.

The authors mentioned in figure 2 legend: 

Results are presented as relative fluorescence units (RFU) and bars express percentages relative to the PBS group. C (-): cells treated with PBS. C (+): cells treated with DNAse (500 μg/mL).

It is unclear whether the assay was conducted in a cell-free or not.

Please, make it clear.

Author Response

RESPONSE TO REVIEWERS

Title: A promising biomolecule able to degrade neutrophil extracellular traps: CdcPDE, a rattlesnake phosphodiesterase

Thank you very much for your considerable effort in reviewing our manuscript. It is also appreciated that you considered our work of interest for your journal and its readers to allow the submission of a revised version. It stimulated us to amend the text to meet your constructive comments. In what follows, you will find a point-by-point list of how we dealt with reviewer comments in blue, and necessary changes are highlighted in the manuscript in red. We hope that this version is now acceptable for publication in Toxins.

Comments of the Reviewers:

Reviewer 2:

The manuscript is important and describes the action of a recently isolated nuclease. The authors showed that this enzyme cleaves DNA from cell-free.

The authors mentioned in figure 2 legend:

Results are presented as relative fluorescence units (RFU) and bars express percentages relative to the PBS group. C (-): cells treated with PBS. C (+): cells treated with DNAse (500 μg/mL).

It is unclear whether the assay was conducted in a cell-free or not.

Please, make it clear. 

Response: Indeed, the assay was performed in a cell-free environment using free DNA. Details were added in the legend.

Reviewer 3 Report

The paper shows that CdcPDE, a nuclease from rattlesnake venom, can degrade cell-free DNA in vitro. It is quite a preliminary observation. While the paper claims to be focussed on NETs, it only assesses degradation of commercially-sourced DNA. As such, the authors should modulate statements on lines 59 and 75 implying that this molecule has now been shown to degrade NETs.

In line 45 and onwards, more effort should be made to describe exactly what assay is being performed and what is being measured.

Line 47 – the enzyme reduced the DNA by 38%, not 62%

It would have been informative to have a dose response curve to compare DNAse and CdcPDE.

Author Response

RESPONSE TO REVIEWERS

Title: A promising biomolecule able to degrade neutrophil extracellular traps: CdcPDE, a rattlesnake phosphodiesterase

Thank you very much for your considerable effort in reviewing our manuscript. It is also appreciated that you considered our work of interest for your journal and its readers to allow the submission of a revised version. It stimulated us to amend the text to meet your constructive comments. In what follows, you will find a point-by-point list of how we dealt with reviewer comments in blue, and necessary changes are highlighted in the manuscript in red. We hope that this version is now acceptable for publication in Toxins.

Comments of the Reviewers:

Reviewer 3:

The paper shows that CdcPDE, a nuclease from rattlesnake venom, can degrade cell-free DNA in vitro. It is quite a preliminary observation. While the paper claims to be focused on NETs, it only assesses degradation of commercially-sourced DNA. As such, the authors should modulate statements on lines 59 and 75 implying that this molecule has now been shown to degrade NETs.

Response: Indeed, this is an important observation. We make sure to make it clear through the manuscript.

In line 45 and onwards, more effort should be made to describe exactly what assay is being performed and what is being measured.

Response: We did that.

Line 47 – the enzyme reduced the DNA by 38%, not 62%

Response: This information was corrected. Sorry for that and thanks for the important correction.

It would have been informative to have a dose response curve to compare DNAse and CdcPDE. 

Response: Unfortunately, we could not test CdcPDE using higher concentration due to the low recovery of CdcPDE in the venom (0.71%); however, we used a lower concentration of CdcPDE and did not see the expected result as presented in the figure 2. Therefore, we believe that concentrations lower than 10 µg/mL will not observe Cell-free DNA degradation.

Reviewer 4 Report

The manuscript no. 2111771, entitled A promising biomolecule able to degrade neutrophil extracellular traps: CdcPDE, a rattlesnake phosphodiesterase, reports the capacity of CdcPE, a nuclease isolated from Crotalus durissus collilineatus to hydrolyze DNA from neutrophil extracellular traps, making it a useful candidate in inflammatory diseases. 

The Communication is well written, easy to follow, and its English is adequate. However, it is the Editor, who should judge whether the results are of adequate novelty to warrant a rapid publication in Toxins. After all, the results of a single test are presented in the manuscript. The main flaw of the study is that higher concentrations of CdcPDE were not tested, due to most probably the lack of sufficient quantities of the enzyme.

Minor corrections:

In Key contributions

            Please delete “As” from line 14. The sentence should start like: “A nuclease from….”

Introduction

            Line 29 - Please delete “indeed”, as it repetitively appears in both sentences (lines 28-29)

            Lines 34-35 – Please correct as follows: “…rattlesnake venom, and was completely characterized.”

            Lines 35-36 – Please correct as follows: “…was able to inhibit”

Figure 1 – legend – please correct as follows: “…to illustrate the mechanism of NET degradation…”

Author Response

RESPONSE TO REVIEWERS

Title: A promising biomolecule able to degrade neutrophil extracellular traps: CdcPDE, a rattlesnake phosphodiesterase

Thank you very much for your considerable effort in reviewing our manuscript. It is also appreciated that you considered our work of interest for your journal and its readers to allow the submission of a revised version. It stimulated us to amend the text to meet your constructive comments. In what follows, you will find a point-by-point list of how we dealt with reviewer comments in blue, and necessary changes are highlighted in the manuscript in red. We hope that this version is now acceptable for publication in Toxins.

Comments of the Reviewers:

Reviewer 4:

The manuscript no. 2111771, entitled A promising biomolecule able to degrade neutrophil extracellular traps: CdcPDE, a rattlesnake phosphodiesterase, reports the capacity of CdcPE, a nuclease isolated from Crotalus durissus collilineatus to hydrolyze DNA from neutrophil extracellular traps, making it a useful candidate in inflammatory diseases.

The Communication is well written, easy to follow, and its English is adequate. However, it is the Editor, who should judge whether the results are of adequate novelty to warrant a rapid publication in Toxins. After all, the results of a single test are presented in the manuscript. The main flaw of the study is that higher concentrations of CdcPDE were not tested, due to most probably the lack of sufficient quantities of the enzyme.

Minor corrections:

In Key contributions

            Please delete “As” from line 14. The sentence should start like: “A nuclease from….”

Introduction

            Line 29 - Please delete “indeed”, as it repetitively appears in both sentences (lines 28-29)

            Lines 34-35 – Please correct as follows: “…rattlesnake venom, and was completely characterized.”

            Lines 35-36 – Please correct as follows: “…was able to inhibit”

Figure 1 – legend – please correct as follows: “…to illustrate the mechanism of NET degradation…”.

Response: All considerations have been done and highlighted in the manuscript.

Round 2

Reviewer 3 Report

The authors have made some minor efforts to improve the manuscript.

Author Response

Response: We made changes based on reviewers and editor feedback. We would be glad to correct and improve the manuscript if reviewer point it out.